# Intrahepatic Cholangiocarcinoma Developing in Patients with Metabolic Syndrome Is Characterized by Osteopontin Overexpression in the Tumor Stroma

**DOI:** 10.3390/ijms24054748

**Published:** 2023-03-01

**Authors:** Massimiliano Cadamuro, Samantha Sarcognato, Riccardo Camerotto, Noemi Girardi, Alberto Lasagni, Giacomo Zanus, Umberto Cillo, Enrico Gringeri, Giovanni Morana, Mario Strazzabosco, Elena Campello, Paolo Simioni, Maria Guido, Luca Fabris

**Affiliations:** 1General Internal Medicine Unit, Padua University-Hospital, 35128 Padua, Italy; 2Department of Medicine—DIMED, University of Padua, 35128 Padua, Italy; 3Department of Pathology, Azienda ULSS2 Marca Trevigiana, 31100 Treviso, Italy; 4Department of Molecular Medicine (DMM), University of Padua, 35128 Padua, Italy; 54th Surgery Unit, Azienda ULSS2 Marca Trevigiana, 31100 Treviso, Italy; 6Department of Surgery, Oncology and Gastroenterology—DISCOG, University of Padova, 35128 Padua, Italy; 7Hepatobiliary Surgery and Liver Transplantation Unit, Padua University-Hospital, 35128 Padua, Italy; 8Division of Radiology, Treviso Regional Hospital, 31100 Treviso, Italy; 9Digestive Disease Section, Liver Center, Yale University, New Haven, CT 06510, USA; 10Thrombotic and Haemorrhagic Disease Unit and Haemophilia Center, Department of Medicine (DIMED), University of Padua, 35128 Padua, Italy

**Keywords:** extracellular matrix, stemness, motility, osteopontin, tenascin C, periostin, CD133, CD44

## Abstract

Metabolic syndrome (MetS) is a common condition closely associated with non-alcoholic fatty liver disease/non-alcoholic steatohepatitis (NAFLD/NASH). Recent meta-analyses show that MetS can be prodromal to intrahepatic cholangiocarcinoma (iCCA) development, a liver tumor with features of biliary differentiation characterized by dense extracellular matrix (ECM) deposition. Since ECM remodeling is a key event in the vascular complications of MetS, we aimed at evaluating whether MetS patients with iCCA present qualitative and quantitative changes in the ECM able to incite biliary tumorigenesis. In 22 iCCAs with MetS undergoing surgical resection, we found a significantly increased deposition of osteopontin (OPN), tenascin C (TnC), and periostin (POSTN) compared to the matched peritumoral areas. Moreover, OPN deposition in MetS iCCAs was also significantly increased when compared to iCCA samples without MetS (non-MetS iCCAs, n = 44). OPN, TnC, and POSTN significantly stimulated cell motility and the cancer-stem-cell-like phenotype in HuCCT-1 (human iCCA cell line). In MetS iCCAs, fibrosis distribution and components differed quantitatively and qualitatively from non-MetS iCCAs. We therefore propose overexpression of OPN as a distinctive trait of MetS iCCA. Since OPN stimulates malignant properties of iCCA cells, it may provide an interesting predictive biomarker and a putative therapeutic target in MetS patients with iCCA.

## 1. Introduction

Cholangiocarcinoma (CCA), a primary liver epithelial cancer that can arise from any tract of the biliary tree, is one of the most aggressive and lethal malignancies worldwide. Anatomically, CCAs are classified as intrahepatic (iCCA), peri-hilar (pCCA), and distal (dCCA) [1]. CCA is a rare tumor, at least in the Western countries, with an incidence ranging between 0.3 and 6:100,000 inhabitants/year depending on the geographical area. However, the global incidence is progressively growing in the recent decades, particularly for iCCA [1,2]. Unfortunately, prognosis has not substantially changed, and remains dismal, with a mortality rate of about 1–6:100,000 inhabitants/year and a 5-year survival of only 5–20% [1].

CCA aggressiveness and propensity to early disseminate is influenced by the dense tumor reactive stroma (TRS), which expands in conjunction with the growth of the malignant epithelial counterpart. TRS is composed by an acellular component, the extracellular matrix (ECM), and by several cell populations, including cancer-associated fibroblasts and a polymorphic inflammatory infiltrate [3,4]. Owing to these features, diagnosis often comes late, and available therapies are of limited efficacy, hampering the drug delivery to the tumoral site [5]. Currently, in CCA, the only curative intervention remains surgical resection or, in selected cases, liver transplantation [6]. Unfortunately, the early spread of CCA to the proximal lymph nodes, occurring in more than 70% of patients and favored by the TRS, often precludes the suitability to curative approaches [7].

Although iCCA often arises in the context of a non-cirrhotic liver, it is thought that the chronicization of the inflammatory response sustained by the hepatic repair mechanisms, as observed in hepatitis C virus (HCV) infection, recurrent acute cholangitis, and primary sclerosing cholangitis, may incite the onset and progression of the tumor [8]. Liver repair is driven by the hepatic reparative/regenerative complex, which is characterized by multiple morphological changes, including the generation of a ductular reaction (DR) and biliary metaplasia of hepatocytes (MHs), in an attempt to restore the normal hepatic homeostasis [9]. Thanks to the ability to secrete a wide range of fibro-inflammatory mediators, encompassing cytokines, chemokines, and growth factors possibly supporting cholangiocarcinogenesis, the DR may hold oncogenic potential [10]. Of note, during biliary tumorigenesis, the normal ECM, mainly composed of collagens (type I to V), fibronectin, laminins, nidogens, and perlecan, is progressively dismantled, and qualitatively modified by a de novo deposition of aberrant matrix proteins. These include periostin (POSTN), tenascin-C (TnC), and osteopontin (OPN), which are not usually secreted in the normal ECM. Interestingly, the expression of these matrix proteins correlates with the tendency to metastasize to the lymph nodes of tumors and a lower overall survival of patients, though the underlying mechanisms are far from being understood [7].

Notably, recent epidemiological studies suggest that metabolic syndrome (MetS), frequently associated with non-alcoholic fatty liver disease (NAFLD) and non-alcoholic steatohepatitis (NASH), may be considered an emerging risk factor for the development of iCCA [11]. NASH, in particular, is characterized by the activation of the hepatic reparative/regenerative mechanisms, resulting in a brisk DR, which correlates with the degree of fibrosis. However, no studies so far have explored the pathophysiological association of MetS with DR, ECM remodeling, and cholangiocarcinogenesis. Of note, ECM remodeling is a key mechanism common to liver repair and cardiovascular complications, which are the leading causes of mortality in NAFLD patients. 

As the prevalence of NAFLD/NASH is steadily increasing, even in young individuals in the Western population [12,13], it is conceivable to expect a rise in iCCA associated with MetS. Therefore, the identification of putative biomarkers related to liver repair mechanisms predicting the development of this insidious form of cancer may be of great help for risk stratification. Starting from these premises, in the present manuscript, we evaluated morphological differences between iCCAs associated or not with MetS with respect to hepatic reparative/regenerative responses and ECM modifications to unveil elements with pro-tumorigenic significance.

## 2. Results

### 2.1. Demographic and Clinical Features of iCCA Patients with and without MetS

According to their metabolic profile, patients with iCCA were divided into two cohorts, with (MetS iCCA, n = 22) and without MetS (non-MetS iCCA, n = 44). Categorization of MetS was based on the evaluation of the five major clinical components, encompassing obesity (body mass index (BMI) > 30 kg/m^2^ or waist to hip ratio >0.90 in males and >0.85 in females), type 2 diabetes mellitus (T2DM), elevated blood pressure, increased serum triglyceride levels, or decreased serum high-density lipoprotein (HDL) cholesterol levels (National Cholesterol Education Program Adult Treatment Panel III (NCEP-ATP III) guidelines) (Table 1), which define MetS when clustered at least in three areas. The two cohorts were comparable in terms of age, sex balance, and serological features, whereas, as expected, metabolic abnormalities were more represented in the MetS cohort, except for HDL levels. There were also no significant statistical differences in the entity of liver steatosis in adjacent liver tissue between the two cohorts. Furthermore, no significant differences were observed in treatment modalities and clinical outcome (follow-up time from 95 to 3233 days), evaluated as overall survival (OS), recurrence, and relapse-free survival (RFS) between the two cohorts (Table 2).

Assessment of visceral adiposity, the hallmark of MetS, was performed by calculating the total volume and the volume rate of fat in the abdominal computerized tomography (CT) scan. Significantly higher values of both measurements were observed in MetS iCCA patients compared with non-MetS iCCA patients (Figure 1A,B), thereby providing further evidence of the different metabolic dysregulation affecting the two groups of patients.

### 2.2. Histological Analysis of Hepatic Steatosis, DR, MHs, and Fibrosis in Liver Tissue Adjacent to iCCA Showed no Significant Differences between Patients with and without MetS, though Fibrosis Patterns Were Different

In the matched peritumoral area of iCCA samples, we performed morphometric analysis to evaluate the presence and extension of hepatic steatosis and the degree and patterns of fibrosis. Moreover, we performed immunostaining for the biliary-specific cytokeratin (K) 7 to evaluate the presence and extension of two elements of the hepatic reparative/regenerative system (DR and MHs). Hepatic steatosis, although higher in MetS than in non-MetS iCCA, did not reach statistical significance (Figure 2). Again, we observe that the extent of the DR and MHs did not differ between the two cohorts of patients (Figure 2). 

However, by Masson’s trichrome staining performed in serial histological sections of those used for immunohistochemical analysis, we found that the presence of fibrosis in MetS iCCAs was overall higher than in non-MetS (88% vs. 70%, respectively), even though there was a similar percentage of cirrhotic livers. Furthermore, in MetS iCCAs, fibrosis showed a significantly increased septal pattern compared to non-MetS iCCAs, with a prominent pericentral distribution consistent with that typically observed in liver disease associated with metabolic dysfunction (Figure 3A–C). 

Taken together, these data indicate that, although without significant differences in the activation of the hepatic reparative/regenerative mechanisms, fibrogenesis behaved differently when iCCA developed in the setting of MetS. Starting from these observations, we turned to the evaluation of the qualitative composition of the fibrotic tissue in both peritumoral and tumoral samples by assessing the expression of three pathological components of the ECM associated with iCCA, namely tenascin (TnC), periostin (POSTN), and osteopontin (OPN) [7].

### 2.3. POSTN, TnC, and OPN Expression Were Significantly Increased in Tumor Compared to Peritumor Areas in Both MetS and Non-MetS iCCA; OPN Was More Expressed in MetS iCCA with Respect to Non-MetS iCCA

By comparing the bulk tumor with the peritumoral area, we found a significant up-regulation of all the three ECM proteins, POSTN, TnC, and OPN, in the TRS in both the MetS and non-MetS iCCA. These observations confirm that newly synthesized ECM components accumulate in conjunction with biliary tumorigenesis in either MetS or non-MetS conditions, likely mediating putative pro-oncogenic effects [7] (Figure 4). However, among the three proteins, OPN showed a marked increase in the TRS of MetS iCCA compared to non-MetS iCCA, whereas POSTN and TnC did not show significant differences (Figure 4). Therefore, we identified up-regulation of OPN as a distinctive feature of iCCA developing in the setting of MetS, implying that OPN overexpression may hold a significance related to the metabolic derangement.

Once the increased expression of POSTN, TnC, and OPN was confirmed in histological sections of iCCA, we next tested their in vitro effects on a range of hallmarks of tumorigenesis, which included cell viability, cell migration, and induction of stemness features, using a human iCCA cell line, i.e., HuCCT-1 [14,15]. 

### 2.4. Treatment with OPN and POSTN but Not with TnC Slightly Sustained Cell Viability in iCCA Cells In Vitro

Treatments with OPN and POSTN but not TnC exerted a small but significant stimulus (*p* < 0.05 vs. untreated controls) on cell viability of the HuCCT-1 cell line (Figure 5). It must be underlined that in line with previous studies describing a pro-proliferation effect of these ECM proteins, our data showed that this effect, though detectable, was not so pronounced.

### 2.5. Treatment with OPN, TnC, and POSTN Potently Stimulated iCCA Cell Motility

Unlike cell viability, OPN, TnC, and POSTN induced a potent time-dependent pro-migratory stimulus on HuCCT-1 cells. Twenty-four-hour treatment with OPN, TnC, and POSTN was nearly able to close the scar produced on cell monolayers for a wound healing assay with a comparable action. This effect was significantly higher as compared to untreated controls (Figure 6). Therefore, these data confirm the role of these deregulated ECM proteins to sustain cell motility mechanisms in malignant intrahepatic cholangiocytes.

### 2.6. Treatment with OPN, TnC, and POSTN Induced iCCA Cells to Acquire Cancer-Stem-Cell-like Phenotypic Traits

An additional effect exerted by cell–ECM interactions occurring in the TRS is the gain of a cancer stem cell (CSC)-like phenotype, which is relevant for tumor initiation, chemoresistance, and tumor recurrence. By assessing the mRNA levels of CD133 and CD44, two well-established markers displayed by CSCs, we found that 24 h treatment of HuCCT-1 cells with OPN, TnC, and POSTN variably modulated their expression of CD133 and CD44, which increased significantly after challenge with OPN and POSTN, but not with TnC (Figure 7). 

Taken together, these data indicate that de novo expression of abnormal ECM components in the TRS of iCCA is functionally relevant to promote motility and CSC features of malignant cholangiocytes, and in this respect, OPN behaves as a key determinant of the enhanced fibrogenesis featuring MetS-associated iCCA.

## 3. Discussion

In recent years, the incidence of CCA has been increasing worldwide, as observed in the Western countries for the iCCA variant. Among the predisposing disease conditions responsible for this heavier epidemiological burden in Europe as well as in the US, MetS represents an emerging risk factor, showing an OR for iCCA of 1.73 in patients with T2DM and 2.2 in patients with NAFLD, the hepatic manifestation of the MetS [1]. Indeed, NAFLD/NASH has become the most widespread chronic liver disease in the Western populations, and its incidence has paralleled the increased diffusion of the MetS [16].

*NAFLD/NASH may have a pro-carcinogenic role in the development of iCCA*. Thus, recent epidemiological evidence clearly indicates that MetS is a risk factor not only for hepatocellular carcinoma, but also for iCCA [11,17]. However, whereas the mechanisms by which MetS and the related liver involvement, NAFLD/NASH, sustain carcinogenesis towards HCC have drawn attention, underlying the role of the long-lasting inflammatory response induced by the lipotoxicity affecting the hepatocytes [18], how a metabolic derangement may induce pro-oncogenic effects on the cholangiocyte level has not been investigated yet. 

Starting from the assumption that fibrogenesis is an important mechanism related to chronic inflammation endowing malignant potential, in this study we aimed to verify whether iCCA developing in patients with MetS shows distinctive alterations of fibrosis and ECM components compared to patients with iCCA without MetS. By focusing on three matrix components typically up-regulated in the tumor microenvironment (TME) of iCCA (POSTN, TnC, and OPN), we also evaluated their effects on the biology of iCCA cells.

In a single-center series of iCCA patients undergoing surgical resection with curative intent, we considered two cohorts on the basis of their association with MetS, defined by the presence of the five key clinical components including obesity, elevated blood pressure, increased serum glucose or triglyceride levels, or decreased serum HDL cholesterol levels. Apart from these metabolic parameters, the two cohorts were well comparable in terms of clinical and demographic features. As confirmation of the altered metabolic profile, abdominal CT scan was also evaluated to assess visceral adiposity (a hallmark of MetS), which resulted as significantly increased visceral fat volume and rate in the MetS-associated iCCA, further supporting the patient categorization.

In patients with iCCA, the presence of MetS is associated with a diverse pattern of fibrosis developing in the peritumoral tissue, whereby activation of the cell elements of the hepatic repair response does not differ. The development of cancer is not an all or nothing phenomenon and usually starts from pre-tumoral lesions, which under a persistent chronic inflammatory stimulus responsible for genomic instability develop into malignant tumors owing to sustained proliferative signaling, resistance to cell death, limitless replicative potential, and loss of growth suppression [8]. In chronic liver diseases, the chronic inflammatory stimulus is driven by the activation of epithelial structures belonging to the hepatic reparative/regenerative system, composed of ductular reactive cells associated with a range of inflammatory cells and myofibroblasts, resulting in a ductular reaction (DR), and in metaplasia of hepatocytes (MHs) [9], which are not present in the normal healthy liver [19]. In this first part of the study, we drew our attention on the matched peritumoral areas of the resected iCCA to evaluate the liver background from which the malignant transformation originated. Although the extent of these aberrant epithelial structures was increased with respect to normal condition [19], no significant differences were found between the two cohorts, indicating the lack of a significant effect of MetS in eliciting a DR/MHs in the liver. Conversely, we found significant differences with respect to the type and extent of fibrosis between the two study groups. In the MetS cohort, we found a significant preponderance of stage 4 (septal) fibrosis, in keeping with the pattern commonly observed in the hepatic metabolic injury [20] and indicating a more advanced fibrotic progression upon metabolic dysregulation. Moreover, our data highlight the concept that MetS behaves as a warning sign of liver fibrosis [21], as suggested by a recent meta-analysis demonstrating that MetS was an independent risk factor for hepatic fibrosis even in the absence of steatosis [22]. Of note, in our study, although the mean percentage of liver steatosis tended to be higher in the group of iCCA with MetS compared with that without MetS (12.05 vs. 5.54%), it did not reach statistical significance.

*ECM in the TRS of iCCA is characterized by de novo deposition of TnC, POSTN, and OPN.* After the evaluation of the peritumoral tissue, we turned to the matched bulk tumor to see if ECM protein accumulation was quantitatively and qualitatively different in the TRS between MetS and non-MetS iCCAs. Although development of an abundant TRS is a distinctive feature of iCCA, no data highlighting a different TRS composition across diverse underlying liver disease etiologies of CCA are available. Among the various components of the TRS, herein we focused on the ECM proteins, starting from the assumption that intensive ECM remodeling occurs in MetS [23] and from our evidence indicating a different fibrotic pattern between the two groups of iCCA. In both cohorts, the deposition of TnC, POSTN, and OPN was significantly higher compared with the peritumoral tissue, whereby they were almost absent. This observation confirms the concept that these ECM proteins are newly secreted in conditions of malignant transformation, implying a more complex functional role beyond the structural rearrangement. Moreover, OPN was significantly more expressed in MetS- than non-MetS iCCA tumor specimens, hinting at the possibility that OPN plays a specific role in the tumorigenesis of iCCA in patients with MetS. Recent studies have uncovered that OPN behaves as a regulator at the cross roads of inflammation, obesity, and diabetes [24]. The fundamental role played by OPN in mediating metabolic responses was addressed using different in vivo models. The comparison between OPN^−/−^ and WT control mice both fed with a high-fat diet (HFD) demonstrated that OPN expression is essential for the early onset of insulin resistance [25]. Moreover, obese mice treated with specific neutralizing antibodies against OPN showed inhibition of chronic inflammation induced by obesity as well as of insulin resistance development [26]. Furthermore, several works unveiled a close relationship of OPN overexpression with the development of T2DM complicated by nephropathy [24]. 

In the liver, different cell types, such as epithelial, endothelial, and immune cells, may secrete OPN in response to chronic injury [27]. In NASH, in vivo and in vitro studies have pinpointed the profibrogenic role of OPN, not only as an ECM protein but also as a cytokine. Upon treatment with the methionine–choline-deficient (MCD) diet (a classical dietary model of NASH), mice showed an up-regulation of OPN in the DR compartment that was accompanied by increased fibrosis in the liver parenchyma. This effect was reduced in OPN^−/−^ mice and was recapitulated and further exacerbated in Patched (Ptc)^+/−^ mice (harboring overactivation of the Hedgehog (Hh) signaling), indicating a combinatorial interaction of OPN with Hh. Interestingly, in NASH patients OPN expression correlated with activation of the Hh pathway and fibrosis stage [28]. In another study, WT mice treated with an HFD developed an inflammatory response mostly mediated by macrophages, as with human NAFLD. Pharmacological inhibition of Smoothened (SMO) (downstream effector of the Hh signaling) with GCD-0449 and LED225, or HFD treatment to nourish Smo-LKO mice, reduced the infiltration of activated macrophages and the expression of pro-inflammatory mediators [29]. Altogether, these observations revealed that up-regulation of OPN is interwoven with an over-activation of the Hh pathway and this pathogenic link is strategic in directing hepatic fibrogenesis in MetS, likely mediated by a macrophage-driven inflammatory response with enhanced ECM remodeling. 

*In vitro, OPN, TnC, and POSTN induce iCCA cells to gain pro-migratory functions and to acquire a stem-cell-like phenotype*. To understand the pro-oncogenic potential of fibrosis associated with the up-regulation of aberrant ECM proteins, we conducted in vitro studies using an established cell line of human iCCA, the HuCCT-1 cells. Previous literature dealing with other tumor contexts highlighted the ability of TnC, POSTN, and OPN to regulate several core roles, which orchestrate tumorigenesis and tumor progression, such as proliferation, cell viability, migration, angiogenesis, pro-invasive pathways, and stemness induction [30]. Starting from these premises, we analyzed iCCA cell responses to in vitro treatments with these three proteins. Whereas impact on cell viability of iCCA cells was very mild, effects on cell motility were very pronounced and of similar magnitude with all of them. Although the molecular mechanisms by which OPN, TnC, and POSTN promote cell invasiveness in iCCA cells are far from clear, a role of integrin α5β1 in triggering the phosphatidylinositol 3-kinases (PI3K)/Akt pathway and amplifying Met and/or Erb signaling in response to POSTN has been proposed [31].

Furthermore, we found that OPN and POSTN but not TnC significantly stimulated HuCCT-1 cells to overexpress the CSC biomarkers, CD44 and CD133. In human mammary epithelial cells, POSTN provided breast cancer cells with a stem-cell-like phenotype [15]. In CCA, the accumulation of OPN, TnC, and POSTN increased the content of CSCs [31]. In particular, OPN acts as a critical regulator of the CSC niche. OPN exerts a recruiting effect on cancer-associated fibroblasts (CAFs), which in iCCA are the most abundant cell population of the TRS and functionally support the expansion of the CSC niche. A further effect of OPN is the stimulation of macrophages with the M1 phenotype into tumor-associated macrophages (TAMs), which, as with CAFs, are involved in the regulation of CSC function. These modulatory effects on CSCs have been also demonstrated in other malignant settings of the gastrointestinal system, as in hepatocellular carcinoma [32] and colorectal carcinoma [33]. 

*In conclusion*, since CSCs preferentially contribute to tumor initiation, in addition to confirming the pro-tumorigenic role of fibrosis, these observations link fibrosis progression with MetS and identify OPN as a key molecular effector of this pathogenetic link. Notably, OPN can be modified in five isoforms due to alternative splicing (OPNa, OPNb, OPNc, OPN4, and OPN5), which are differentially expressed in different tumors [34,35]. To date, the functions of these splicing variants are not completely clear, and no studies have been conducted on this topic in iCCA, an issue worth being pursued by future research directions. However, since OPN may be also found as a secreted cytokine in biological fluids, including serum, in theory, OPN may provide a tool to monitor the malignant potential of liver fibrosis associated with MetS. In line with this observation, using OPN dosage in serum as a non-invasive biomarker is supported by extensive evidence generated in pancreatic ductal adenocarcinoma and ovarian carcinoma, showing its suitability to evaluate tumor progression and to predict post-operative complications, as found in colorectal cancer [36,37]. 

## 4. Methods and Materials

**Patient selection and clinical data.** A total of 66 (n = 22 with MetS and n = 44 without MetS) consecutive patients diagnosed with primary iCCA who underwent percutaneous biopsy or laparoscopic liver resection from January 2006 to September 2020 were retrospectively included in this study. The exclusion criteria were as follows: neoadjuvant treatment, both systemic and/or locoregional; less than 3 months survival after surgery; and the absence of sufficient materials for additional immunohistochemical analyses. The inclusion criterion was the diagnosis of iCCA (small and large duct types) according to WHO classification 2019 [38]. Clinical and laboratory data, including gender, age, BMI, diagnosis of hypertension and diabetes, fasting blood glucose, percentage of glycated hemoglobin, serum triglyceride and HDL levels, and microalbuminuria and serum levels of alpha-fetoprotein (AFP), carcinoembryonic antigen (CEA), carbohydrate antigen 19.9 (CA19.9) and total bilirubin at the time of surgery, were retrieved from medical records. The presence of any chronic liver or biliary disease, cirrhosis, and any adjuvant, systemic, and/or locoregional treatment was also recorded. For all patients, abdominal pre-surgery CT scans with staging ware recovered. Patients were followed up regularly by measuring serum tumor marker levels and performing CT to detect recurrence of the disease. 

**CT scan evaluation.** Analyses of “picture archiving and communication system” (PACS) were performed by using Somatom AS+ system (Siemens Healthineers, Erlangen, Germany). Acquisitions were performed at 100–120 kV with a pitch of 0.9–1.2 and slice thickness of 2 mm with 1 mm increment and automatic tube current modulation (unenhanced scans). Injected contrast medium, if necessary, was Accupaque 350 (GE Healthcare, Cork, Ireland). Complete abdominal scan from diaphragmatic pillars to small pelvis was performed. To avoid bias caused by biopsies or surgery, CT was performed before surgical intervention. Visceral fat volume measurements were performed by using Synapse 3D (Version 5.5.002, Fujifilm Corporation, Tokyo, Japan). The regions of interest (ROI), identifying the fatty tissue, were individuated by adjusting the density intervals from −200 to −50 HU. 

**Histology and histological evaluation**. All the slides were stained with hematoxylin–eosin and Masson’s trichrome (to evaluate fibrosis) and were revised in double-blinded methods by skilled pathologists specialized in liver diseases (S.S. and M.G.). All cases were classified according to the latest edition of the WHO classification of digestive system tumors (2019) [38]. Relevant histological features were recorded, including grade of tumor differentiation, T stage (according to the revised 8th edition of the UICC staging system) [20], margin status (for surgical resections), the presence of vascular and perineural invasion and lymph node metastasis, and the presence and extent of steatosis, nuclear glycogenosis, and hepatocyte ballooning in the adjacent liver. Considering all the different etiologies of liver disease included in the study, we evaluated fibrosis in a qualitative manner, as follows: 0, no fibrosis; 1, sinusoidal and perivenular (central) fibrosis only; 2, portal fibrosis only; 3, septal fibrosis; 4, cirrhosis.

**Immunohistochemistry**. Tissue microarrays (TMAs) made of formalin-fixed paraffin-embedded iCCA tissue cores (with a diameter of 4 mm) were obtained by selecting two or three representative areas of tumor and adjacent liver tissue from each liver resection case, depending on tumor dimension. All of the samples were processed by using the TMA Master platform (3DHistech, Budapest, Hungary), a semi-automatic and computer-assisted TMA platform. Immunostains were performed on TMA and liver biopsy sections as follows: briefly, sections were deparaffinized in xylene (Carlo Erba, Milan, Italy) and rehydrated with absolute ethanol (Carlo Erba). Endogenous peroxidase activity was blocked by incubating for 15 min in methanol (Sigma-Aldrich, St. Louis, MO, USA) + 10% hydrogen peroxide (Scharlau, Barcelona, Spain). Following appropriate antigen retrieval (a.r.), sections were washed with 0.05% PBS + tween 20 (PBST, both Sigma-Aldrich) and incubated for 10 min at room temperature with Ultra Vision protein block (Thermo Scientific, Waltham, MA, USA) to inhibit non-specific reactions. Then, slides were incubated for 1 h at room temperature with the following antibodies: anti-K7 (clone OV-TL 12/30 clone, Cell Marque; mouse monoclonal; working dilution 1:200; a.r. citrate pH6), anti-POSTN (Abcam, Cambridge, UK; rabbit polyclonal; working dilution 1:100; a.r. tris-EDTA pH 9.0), anti-TnC (Abcam, Cambridge, UK; rabbit polyclonal; working dilution 1:500; a.r. tris-EDTA pH 9.0), and anti-OPN (Abcam, Cambridge, UK; rabbit polyclonal; working dilution 1:1000; no a.r.). Sections were then washed with PBST and incubated for 30 min at room temperature with the appropriate conjugated HRP secondary antibody (EnVision, Agilent, Santa Clara, CA, USA). Slides were developed using 3,3’-diaminobenzedine tetrahydrochloride (DAB, Abcam, Cambridge, UK), counterstained with Gill’s Hematoxylin n°2 (Sigma-Aldrich, St. Louis, MO, USA) and mounted with Eukitt (Bio-Optica, Bologna, Italy).

**Immunohistochemical evaluations of DR, MHs, and ECM proteins.** Presence and extent of DR was semi-quantitatively assessed by evaluating K7 staining by two experienced pathologists (S.S. and M.G.), as follows: 0, no DR; 1, DR in less than 50% of portal tracts (PTs); 2, DR in at least 50% of PTs; 3, DR in at least 50% of PTs, with ductular buds extending into the peri-portal acinar parenchyma. The presence of MHs was evaluated as follows: 0, no MHs; 1, MHs around less than 50% of PTs; 2, MHs around at least 50% of PTs. The extent of matrix proteins (POSTN, TnC, and OPN) was evaluated as follows: 0, absence of staining; 1, focal staining, <25% of the stroma; 2, mild staining, between 25 and 50% of the stroma; 3, diffuse staining, over 50% of the stroma.

**Cell cultures.** Human established CCA cell line HuCCT-1 (Health Science Research Resource Bank, HSRRB, Osaka, Japan) was grown by using RPMI 1640 supplemented with 10% FBS and 1% penicillin (all from Thermo Scientific, Waltham, MA, USA) at 37 °C in a 5% CO_2_ atmosphere. Mycoplasma contamination was excluded by using a specific biochemical test (Lonza, Basel, Switzerland). 

**Cell viability (MTS).** Cell viability was evaluated by MTS (Promega, Madison, WI, USA) assay as suggested by the provider. To achieve that, 5 × 10^3^ cells were seeded in a 96-well plate (Falcon, Glendale, AR, USA), starved for 24 h, and treated with medium supplemented with OPN (50 μg/mL, Prospec, Ness-Ziona, Israel) [39], POSTN (100 ng/mL, Sino Biological, Beijing, China) [31], and TnC (5 μg/mL, R&D Systems, Minneapolis, MN, USA) [40].

**Cell migration (wound healing) assay.** CCA cells were seeded in a 6-well plate, grown until confluence, and then starved for 24 h. Cells were treated as described above with OPN, POSTN, and TnC, and untreated cells were used as control. Each cell monolayer was scratched three times with a sterile p200 tip, and three micrographs were taken at t = 0 h for each wound. Then, on the same scratched area, micrographs were taken again at 2 h, 6 h, and 24 h to measure the area covered by the scratch, by using ImageJ software (NCBI). Values are expressed by normalizing each time point to t = 0 h.

**Real-Time PCR.** Total RNA was extracted from HuCCT-1 by using TRIzol reagent, according to the manufacturer’s instructions (Thermo Scientific, Waltham, MA, USA). The expression levels of the mRNAs were determined by real-time PCR (Rotor-Gene Q, Qiagen, Venlo, The Netherlands), using TaqMan^®^Gene Expression Assay and the predesigned primers for CD133 and CD44. GAPDH was used as a housekeeping gene (all dyes were purchased by Thermo Scientific, Waltham, MA, USA).

***Statistical analysis.*** Statistical comparisons were made using the 2-tailed Student’s T-test, χ^2^ test, or one-way Analysis of Variance (ANOVA) test when necessary, using Origin 2022b (OriginLab Corporation, Northampton, MA, USA). A *p*-value < 0.05 was considered to be significant. 

## Figures and Tables

**Figure 1 ijms-24-04748-f001:**
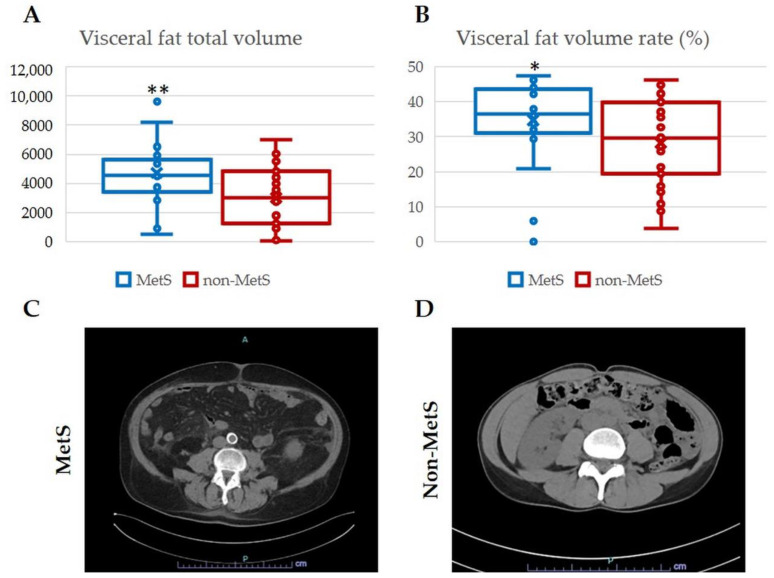
Both visceral fat total volume and visceral fat volume rate were increased in patients with MetS. The volume of visceral fat (**A**) and its percentage (**B**) were significantly increased in MetS with respect to non-MetS iCCA patients. (**C**,**D**) Representative CT scan images of MetS and non-MetS iCCA showing different visceral fat accumulation between the two groups. Measurement was performed by abdominal CT scan. * *p* < 0.05 vs. non-MetS; ** *p* < 0.01 vs. non-MetS using 2-tailed *t* test.

**Figure 2 ijms-24-04748-f002:**
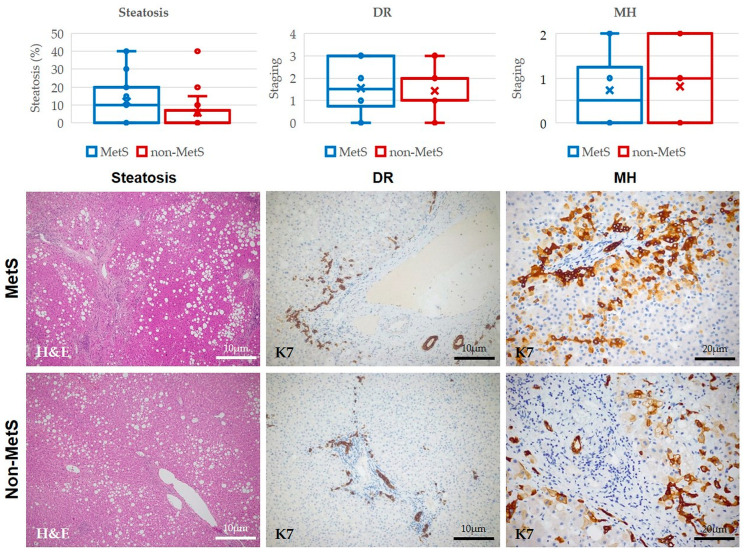
Presence and extent of steatosis, DR, and MHs in the peritumoral tissue of iCCA did not differ between MetS and non-MetS patients. Hematoxylin & Eosin (H&E) histological analysis of liver samples did not show significant differences in the % of steatosis between MetS and non-MetS peri-iCCA tissue. Similarly, by immunohistochemistry for cytokeratin 7 (K7), no significant differences in DR and MHs presence and extent were observed between MetS and non-MetS peri-iCCA tissue. Representative micrographs of peritumoral areas stained for H&E and K7 taken from surgical specimens of MetS and non-MetS patients are reported below the box & whiskers plots. *p* = N.S. (not significant) using 2-tailed *t*-test.

**Figure 3 ijms-24-04748-f003:**
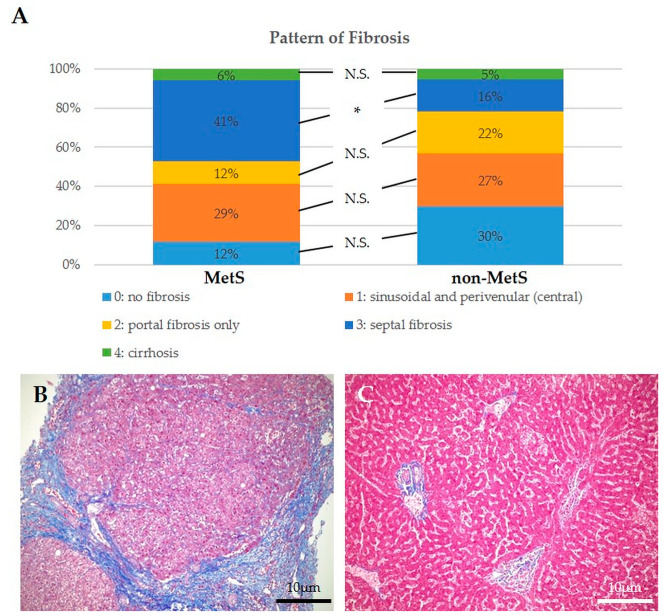
MetS and non-MetS peri-iCCA tissue showed different patterns of hepatic fibrosis. (**A**) The extent and pattern of fibrosis is different between the two groups. A greater proportion of MetS iCCAs showed fibrosis deposition, particularly septal, as compared to non-MetS iCCAs. (**B**) Representative micrographs of cirrhotic progression in a patient with MetS and hepatic steatosis. (**C**) Representative micrograph of a non-MetS iCCA lacking fibrosis. Tissue specimens were taken from peritumoral areas. Staining: Masson’s trichrome. * *p* < 0.05 using χ^2^ test; N.S., not significant.

**Figure 4 ijms-24-04748-f004:**
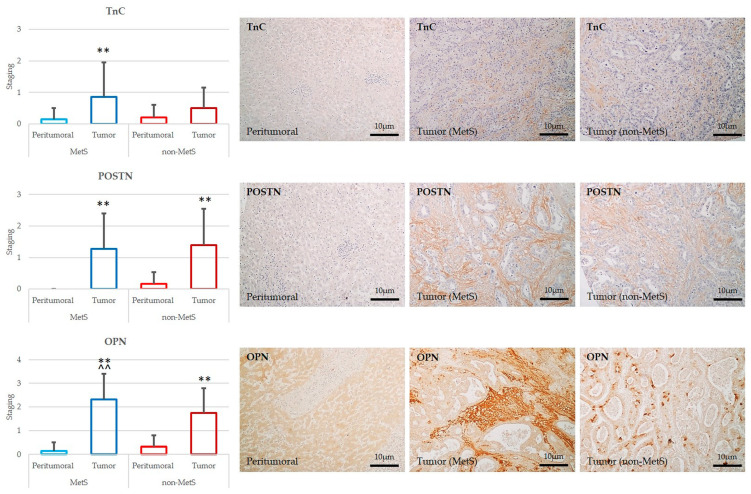
POSTN, TnC, and OPN were more expressed in tumoral than in peritumoral areas of iCCA in either MetS or non-MetS, and OPN overexpression characterized MetS iCCA. The left-sided column plots showed that the deposition of all the three ECM proteins was increased in the bulk tumor compared to the matched peritumoral area in both MetS (n = 22) and non-MetS (n = 44) patients. Notably, OPN overexpression in the TRS discriminated MetS from non-MetS iCCA. ** *p* < 0.01 vs. relative peritumors; ^^ *p* < 0.01 vs. non-MetS iCCA using one-way Analysis of Variance (ANOVA) test.

**Figure 5 ijms-24-04748-f005:**
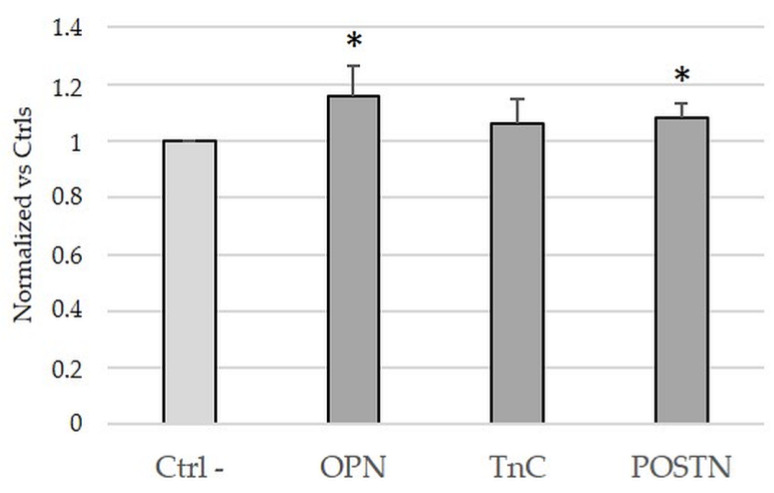
Cell viability of HuCCT-1 cells upon treatment with medium supplemented with OPN, TnC and POSTN. Twenty-four-hour treatment of HuCCT-1 cells with OPN and POSTN, but not with TnC, induced slight increase in cell viability; n = 4 to 6 in duplicate for each experiment. * *p* < 0.01 vs. controls (untreated) using one-way ANOVA test.

**Figure 6 ijms-24-04748-f006:**
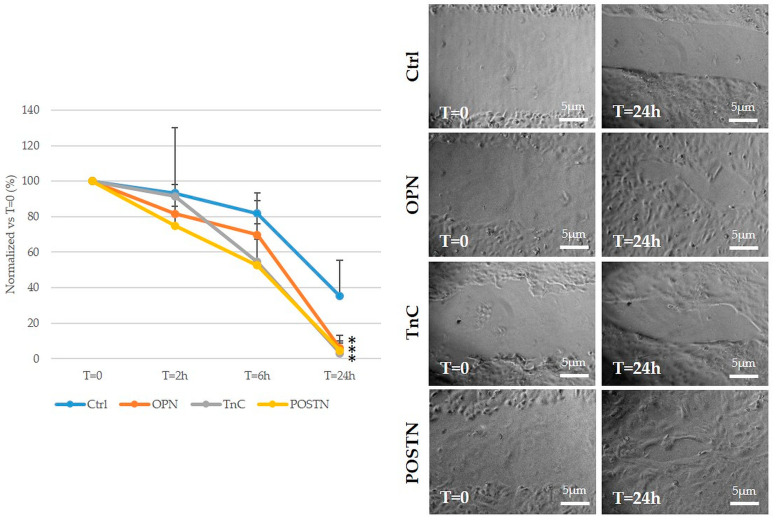
OPN, TnC, and POSTN stimulated migration of HuCCT-1 cells. The in vitro treatment of HuCCT-1 cells with OPN, TnC, and POSTN exerted a potent pro-migratory stimulus on neoplastic cells supporting the pro-invasive properties of abnormally produced ECM in the TRS of iCCA. N = 6; * *p* < 0.01 vs. respective controls (untreated) using one-way ANOVA test. Representative micrographs at T = 0 and T = 24 h.

**Figure 7 ijms-24-04748-f007:**
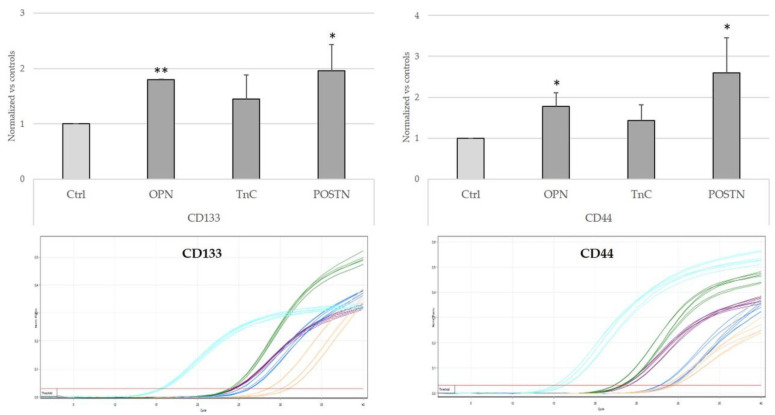
OPN and POSTN but not TnC stimulated expression of the CSC markers CD133 and CD44 of HuCCT-1 cells. A 24 h treatment with OPN and POSTN but not with TnC stimulated cultured iCCA cells to overexpress CD133 and CD44, consistent with a CSC phenotype. N = 3 in duplicate. Below-sided, representative real-time PCR sigmoidal curves for the evaluation of CD133 and CD44 expression (turquoise, housekeeping; yellow, ctrl; violet, OPN; blue, TnC; green, POSTN) are shown. * *p* < 0.05 and ** *p* < 0.01 vs. respective controls (untreated) using one-way ANOVA test.

**Table 1 ijms-24-04748-t001:** Distribution of metabolic abnormalities in MetS vs. non-MetS iCCA patients.

	MetS	Non-MetS	*p*-Value
	22	44	
**BMI > 30 or waist to hip ratio >0.9 (male) or >0.85 (female) (n (%))**	10/18 (55.56)	2/43 (4.65)	<0.001
**T2DM [n (%)]**	18/22 (81.82)	13/34 (38.24)	<0.001
**Systemic hypertension (n (%))**	20/22 (90.90)	13/43 (30.23)	<0.001
**Hypertriglyceridemia (n (%))**	14/19 (73.68)	6/21 (28.57)	0.004
**Low HDL cholesterol (n (%))**	6/14 (42.86)	6/16 (37.50)	0.775

Legend: BMI, body mass index; T2DM, type 2 diabetes mellitus; HDL, high-density lipoprotein.

**Table 2 ijms-24-04748-t002:** Demographic and clinical characteristics of iCCA with and without MetS.

	MetS	Non-MetS	*p*-Value
**N**	22	44	
**Age**	68.36	63.07	0.113
**Female sex (n (%))**	7/22 (31.82)	25/44 (56.82)	0.057
**AFP (ng/mL)**	7.84	83.67	0.488
**CEA (ng/mL)**	11.14	11.60	1
**CA19.9 (U/mL)**	655.01	1121.317	0.1
**CT adjuvant (n (%))**	5/15 (33.33)	16/32 (50.00)	0.294
**Deceased (n (%))**	13/21 (61.90)	26/43 (60.47)	0.913
**OS (days)**	1152.14	937.0732	0.29
**Recurrence (n (%))**	8/21 (38.10)	21/43 (48.84)	0.426
**Relapse-free interval (days)**	869.48	613.51	0.17

Legend: N, number; AFP, alpha-fetoprotein; CEA, carcinoembryonic antigen; CA19.9, carbohydrate antigen 19.9; CT, computerized tomography; OS, overall survival.

## Data Availability

The data that support the findings of this study are available from the corresponding author upon reasonable request.

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
