# Peer review of "Intrahepatic Cholangiocarcinoma Developing in Patients with Metabolic Syndrome Is Characterized by Osteopontin Overexpression in the Tumor Stroma"

_ijms, 2023, doi:10.3390/ijms24054748_

Round 1

Reviewer 1 Report

Dear Author

The manuscript represents an interesting topic for understanding cancer evolution and evaluation.

The arguments presented in the introduction are clearly expressed and put the reader, even if not an expert, in a position to understand the importance of the topic.

The results effectively express the correlation between the rearrangement of the extracellular matrix inside the tumor and outside it.

These data evidence the importance of Osteopontin and Periostin in promoting motility and aggressive cellular disease.

The discussion presents a clear commentary and a good argumentation.

I suggest the authors clarify the role of Osteopontin in many different cancer types, as indicated by other manuscripts (https://pubmed.ncbi.nlm.nih.gov/?term=%22serum+osteopontin%22+cancer&format=abstract&sort=date&size=50).

In conclusion, the manuscript is a very interesting topic.

Author Response

On behalf of the authors, I sincerely thank reviewer 1 for the appreciation of the manuscript and for suggestions to improve the conclusions of the paper.

Point 1: I suggest the authors clarify the role of Osteopontin in many different cancer types, as indicated by other manuscripts (https://pubmed.ncbi.nlm.nih.gov/?term=%22serum+osteopontin%22+cancer&format=abstract&sort=date&size=50).

Response 1: Thank you for your suggestion; a comment on the prospects for using serum osteopontin quantification as a non-invasive marker of outcome in cancer and associated pathologies has been added to page 12, line 384 The suggested manuscript, together with a review have been added as a reference [36] and [37].

Reviewer 2 Report

The article „Intrahepatic cholangiocarcinoma developing in patients with metabolic syndrome is characterized by Osteopontin overexpression in the tumor stroma” by Cadamuro et al. respectably reports and highlights the overexpression of osteopontin (OPN) as a distinctive trait of intrahepatic cholangiocarcinoma in association with metabolic syndrome.

Overall, the information presented represents valuable information regarding OPN as a putative predictive marker and therapeutic target for patients with MetS iCCA. The paper is generally well written in good English and well structured. Conducted experiments and drawn conclusions are comprehensible.  

Apart from minor spelling errors such as missing blank characters before units and lack of explanation of abbreviations I have no complaints.

For more in-depth analysis I recommend performing some gene expression analysis in iCCA patient samples for OPN, TnC and POSTN genes using quantitative Real-time PCR to confirm immunohistochemical findings.  

Minor issues:

1.    Page 9, line 259: What does “TME” stand for? Please clarify in the text.

2.    Page 10, line 305: Please correct “This observation confirms…”.

Author Response

We would like to thank the reviewer for the positive remarks on our manuscript

Point 1: Apart from minor spelling errors such as missing blank characters before units and lack of explanation of abbreviations I have no complaints.

Response 1: I apologize for spelling errors and typos along the text. The manuscript has been thoroughly revised and missing abbreviations were translitterated in extenso.

Point 2: For more in-depth analysis I recommend performing some gene expression analysis in iCCA patient samples for OPN, TnC and POSTN genes using quantitative Real-time PCR to confirm immunohistochemical findings.

Response 2: Thanks to the rewiever for this important suggestion. We are aware that this experiment would give strong support to our immunohistochemical data, but, given the 10 days accounted for the response, we are not able in this short time frame to isolate nucleic acid from all the tumor and peritumor sections, retrotranscribe them, run rt-PCR and analyze the results. Starting from this excellent observation, we are submitting applications to raise founds for these experiments and eventually to expand these observations taking advantage of an european repository of CCA samples (ENS-CCA), in which we are actively involved.

Minor issues

Point 1: Page 9, line 259: What does “TME” stand for? Please clarify in the text.

Response 1: Sorry for this forgetfulness. We have double checked all the acronyms along the text and spelt them out.   

Point 2: Page 10, line 305: Please correct “This observation confirms…”.

Response 1: Sorry again tor this mistake. Text has been thorougly revised to fix some issues and errors.

Reviewer 3 Report

Major points       
1. I miss a more detailed analysis of the several identical domains in the three OPNs such as OPN-a, OPN-b, and OPN-c (alternative splicing isoforms) overexpression in the tumor stroma.
2. Authors should provide a good graph of mRNA expression by real-time PCR in the article so that the rtPCR cycle is displayed.
3. It is better to discuss in detail that metabolic variables such as age, type 2 diabetes, etc., play a role in the positive regulation of OPN.

Minor points

1. Several spelling and grammatical errors in manuscript such as line 32-33  we found a significant increased” or line 361 "liver fibrosis associated to MetS Improve grammar"

2. Improve writing structure.

2. Improve writing structure.

Author Response

We would like to thank the reviewer for the valuable suggestions which have enabled us to improve the quality of the manuscript.

Point 1: I miss a more detailed analysis of the several identical domains in the three OPNs such as OPN-a, OPN-b, and OPN-c (alternative splicing isoforms) overexpression in the tumor stroma.

Response 1: Thank you for rasing this important concept. To address this issue is of paramount importance due to the putative diverse biological effects exerted by the different isoforms of OPN. Unfortunately, the polyclonal antibody used for immunostaining experiments recognizes all the different isoforms, so it does not fit for their discrimination. Moreover, to the best of our knowledge, although there are several anti-OPN antibodies, no one is certified for the specific recognition of the different isoforms in paraffine material. However, to underline the relevance of this observation, comments on this issue have been added in the “conclusion” section, page 11, line 375 and supported by new references [34,35].

Point 2: Authors should provide a good graph of mRNA expression by real-time PCR in the article so that the rtPCR cycle is displayed.

Response 2: To address this observation, two more panels with the sigmoidal graphs of the rt-PCR runs (CD133 and CD44, respectively) have been added to figure 7. The relative figure caption has been modified accordingly.

Point 3: It is better to discuss in detail that metabolic variables such as age, type 2 diabetes, etc., play a role in the positive regulation of OPN.

Response 2: Thanks to the reviewer for this valuable suggestion. Comments regarding the involvement of metabolic variables in human and animal models in the modulaion of OPN expression were added to the “discussion” section, page 10, lines 321-326, and expanded the reference list [25,26].

Minor points

Point 1: Several spelling and grammatical errors in manuscript such as line 32-33  “we found a significant increased” or line 361 "liver fibrosis associated to MetS Improve grammar".

Response 1: Sorry for these mistakes. We have amended these errors and revised all the text to improve grammar and reduce typos.   

Point 2: Improve writing structure.

Response 2: Sorry again tor this issues. Text has been thorougly revised and amended. We do hope that writing structure looks now better.

Round 2

Reviewer 3 Report

- The manuscript still needs extensive revision for language and grammar,